# The Effects of Creatine Monohydrate Loading on Exercise Recovery in Active Women throughout the Menstrual Cycle

**DOI:** 10.3390/nu15163567

**Published:** 2023-08-13

**Authors:** Amanda N. Gordon, Sam R. Moore, Noah D. Patterson, Maggie E. Hostetter, Hannah E. Cabre, Katie R. Hirsch, Anthony C. Hackney, Abbie E. Smith-Ryan

**Affiliations:** 1Applied Physiology Laboratory, Department of Exercise and Sport Science, University of North Carolina at Chapel Hill, Chapel Hill, NC 27599, USA; amandagordon@comcast.net (A.N.G.); smoore@unc.edu (S.R.M.); noah.patterson@colorado.edu (N.D.P.); maggiehostetter99@gmail.com (M.E.H.); hannah.cabre@pbrc.edu (H.E.C.); thackney@med.unc.edu (A.C.H.); 2Human Movement Science Curriculum, Department of Health Sciences, University of North Carolina at Chapel Hill, Chapel Hill, NC 27599, USA; 3Department of Exercise Science, Arnold School of Public Health, University of South Carolina, Columbia, SC 29208, USA; khirsch@mailbox.sc.edu

**Keywords:** dietary supplement, female physiology, menstrual cycle

## Abstract

Creatine supplementation improves anaerobic performance and recovery; however, to date, these outcomes have not been well explored in females. This study evaluated the effect of creatine monohydrate loading on exercise recovery, measured by heart rate variability (HRV) and repeated sprint performance, in women across the menstrual cycle. In this randomized, double-blind, cross-over study, 39 women (mean ± standard deviation: age: 24.6 ± 5.9 years, height: 172.5 ± 42.3 cm, weight: 65.1 ± 8.1 kg, BF: 27.4 ± 5.8%) were randomized to a creatine monohydrate (*n* = 19; 20 g per day in 4 × 5 g doses) or non-caloric PL group (*n* = 20). HRV was measured at rest and after participants completed a repeated sprint cycling test (10 × 6 s maximal sprints). Measurements were conducted before and after supplementation in the follicular/low hormone and luteal/high hormone phases. Creatine monohydrate supplementation did not influence HRV values, as no significant differences were seen in HRV values at rest or postexercise. For repeated sprint outcomes, there was a significant phase × supplement interaction (*p* = 0.048) for fatigue index, with the greatest improvement seen in high hormone in the creatine monohydrate group (−5.8 ± 19.0%) compared to changes in the PL group (0.1 ± 8.1%). Sprint performance and recovery were reduced by the high hormone for both groups. Though not statistically significant, the data suggests that creatine monohydrate could help counteract performance decrements caused by the high hormone. This data can help inform creatine monohydrate loading strategies for females, demonstrating potential benefits in the high hormone phase.

## 1. Introduction

Creatine is regarded as one of the most effective ergogenic aids for exercise performance [1]. More recently, data suggest that creatine supplementation may be particularly beneficial in females due to hormonally influenced fluctuations in creatine kinase activity and differential baseline levels of phosphocreatine (PCr) between males and females [2]. With lower levels of PCr reported in women, this may result in greater performance benefits for females with creatine supplementation. Reports indicate that females are the largest group of dietary supplement consumers [3], emphasizing the need to better understand the sex-specific effects of dietary supplements and the role of female physiology.

As a result of improving cellular metabolism, creatine monohydrate supplementation has led to improvements in anaerobic exercise performance and recovery by helping to maintain pH [4], augmenting glycogen storage [5], and decreasing inflammation [6]. A main contributor to skeletal muscle metabolism during exercise and exercise recovery is the depletion and replenishment of muscle glycogen. Males and females exhibit similar increases in total PCr post-supplementation, while females have exhibited higher levels of intramuscular PCr, suggesting sex-divergent responses to creatine monohydrate supplementation [7]. Though creatine monohydrate supplementation improves mechanisms implicated in accelerated exercise recovery, little research has evaluated the effects on acute post-exercise recovery. Creatine monohydrate supplementation has the potential to influence recovery by increasing intramuscular PCr concentrations, helping to maintain pH, and augmenting glycogen storage [1,2,4,8,9,10,11]. Short-term creatine supplementation is widely associated with improved anaerobic exercise. Creatine monohydrate supplementation has resulted in improvements in average power and peak power over 15-s sprints in trained males [12], with similar results in females [13], but lacking control for the menstrual cycle.

Exercise recovery can be characterized as HRV, represented by the balance and fluctuation between the parasympathetic and sympathetic nervous systems (autonomic function) [14]. A higher overall HRV is an indicator of self-regulatory capacity and efficiency [15]. Exercise has been shown to positively influence HRV by increasing the time needed for successive RR intervals to return to baseline. In healthy individuals, HRV was shown to significantly increase at rest before exercise as well as during a recovery period after exercise following a ten-week aerobic training program [16]. Existing data report small variations in HRV across the menstrual cycle, with a slight decrease, or no change [17], in the luteal phase (high hormone) compared to the follicular phase (low hormone) [18,19]. After exercise, autonomic activity heavily influences the restoration of homeostasis; the sympathetic nervous system is dominant in allocating energy where it is most necessary [20]. If creatine increases the efficiency of the body to elicit the recovery process, as reflected by parasympathetic drive, HRV values such as the root mean square of successive differences (RMSSD) or the standard deviation of normal-to-normal intervals (SDNN) could see significant changes post-exercise and post-supplementation; these mechanisms can also be influenced by the menstrual cycle. Creatine metabolism could potentially vary based on the menstrual phase due to the suppressive effect of estrogen on glycolytic enzymes, fluctuation in creatine kinase (CK), as well as the effect of estrogen levels on the rate-limiting enzyme in creatine synthesis, arginine-glycine aminotransferase [2,7,21].

To date, the mechanistic potential for creatine to enhance recovery has yet to be evaluated in women, particularly when controlled for the menstrual cycle [2,7]. Due to the significant physiological differences between women and men, particularly related to the menstrual cycle and exogenous hormones (i.e., oral contraceptives), it is expected that exercise recovery will be affected [21]. To our knowledge, HRV response has yet to be examined during exercise recovery in different menstrual phases with creatine monohydrate supplementation. This study aimed to assess the effect of creatine monohydrate loading on exercise recovery measured from HRV (SDNN, RMSSD) and repeated sprint outcomes (average power, peak power, fatigue index, time to peak power) in women across menstrual phases.

## 2. Materials and Methods

### 2.1. Subjects

Fifty-two women were enrolled in this study; thirteen dropped out due to time constraints after randomization and became ineligible after enrolling (Figure 1). Data were analyzed for 39 women (Table 1) randomized to either the creatine monohydrate group (*n* = 19) or the PL group (*n* = 20). Participants who were naturally menstruating (*n* = 25), using birth control monophasic oral contraceptives (*n* = 8), or intrauterine devices [IUD; hormonal and copper] and vaginal rings (*n* = 6) were included (Figure 1).

Sample size estimations were determined a priori based on the primary outcome variables (HRV, fatigue index, and power output), with an average effect size of 0.50, correlation of 0.5, 80% power, and an alpha level of *p* = 0.05, requiring a total sample size of 42 (*n* = 21 per treatment). Participants who completed at least one phase (low hormone or high hormone) of pre- and post-supplementation visits were included in the analysis (*n* = 39). Participants enrolled were recreationally active (exercising at least 3 days a week) and normally menstruating, with a BMI between 18.5 and 29.9 kg/m^2^. Despite the wide inclusion, the average BMI was 23.7 ± 2.7 kg/m^2^, with an average percent body fat of 27.4 ± 5.8% (<25% percentile per NHANES), representing a group of normal-weight, lean women. Women were excluded if they were amenorrheic, using a biphasic or triphasic oral contraceptive, had changed hormonal contraceptive type within the previous six months, used prescription medication (besides hormonal contraception), consumed creatine or other ergogenic aids, were pregnant, or had experienced a musculoskeletal injury in the previous three months. Women were either novice creatine users or had not consumed creatine supplementation within the previous six months. All procedures in this study were approved by the University’s institutional review board in accordance with the Declaration of Helsinki, and all participants provided written informed consent before enrollment. 

### 2.2. Experimental Design

This study had a randomized, double-blind, placebo-controlled design and included five laboratory visits. Participants were randomly assigned to treatment order and menstrual cycle phase order using a computer-generated allocation sequence (Sealed Envelope Ltd., London, UK, 2022). Each visit was conducted following a minimum of an eight-hour fast and having refrained from vigorous exercise for at least 48 h, and from caffeine and alcohol for 24 h before each visit. Subjects were asked to maintain their normal diet pattern throughout their participation but were asked to consume no more than 200 mg of caffeine per day. A urine sample was taken to confirm non-pregnant status (HCG test) and hydration status (1.002–1.025) via urine-specific gravity. Anthropometric measures were collected, followed by a 30-min resting HRV measure, a body composition measurement from dual-energy x-ray absorptiometry, a repeated sprint ability test, and a 15-min recovery test measuring HRV. 

A familiarization session was completed at the start of their menstrual cycle (days 0–5) in order to introduce the protocol. Participants were then randomly assigned to return in either their low hormone phase (beginning on days 2–8) or high hormone phase (beginning on days 14–18) and were randomly assigned to either a creatine monohydrate (20 g/day of creatine monohydrate for five days) or placebo (PL, 20 g/day of non-caloric placebo for five days) group. The low hormone phase for the participants on oral contraceptives was defined as their placebo or withdrawal pills, and the high hormone phase was considered the three weeks of active pills. Participants completed a minimum four-week washout period between their supplementation periods.

### 2.3. Phase Tracking

Participants with naturally occurring menstrual cycles (not on hormonal birth control) tracked their menstrual cycle by recording daily basal body temperature and menstrual symptoms into a tracking app (FertilityFriend) accessible to both the participant and researcher. This data was used to indicate cycle start date, cycle length, and ovulation date, and was used to confirm status prior to testing sessions. Salivary estrogen was also used to determine the cycle phase. Estrogen levels were determined from a 2.5 mL passive drool salvia sample analyzed by an ELISA assay (Salivary 17 β-Estradiol Enzyme Immunoassay Kit, Salimetrics, LLC, State College, PA, USA). 

### 2.4. Heart Rate Variability

Heart rate variability was measured using a heart rate monitor strapped to the participant’s chest (Polar H10, Polar Electro Oy, Kempele, Finland) and assessed through the EliteHRV app for SDNN and RMSSD. At the beginning of each visit, participants were asked to lie supine on a table for 30-min, while wearing a chest heart rate monitor (Polar H10, Polar Electro Oy) to obtain a resting HRV recording. Standard deviation of normal-to-normal intervals (SDNN) and RMSSD were assessed through an app (EliteHRV, Gloucester, MA, USA). After completing the repeated sprint protocol, a 15-min recovery HRV reading was taken in five-minute increments (5 min, 10 min, 15 min) and averaged to determine recovery; the average of the first 5 min of recovery was also defined as immediate recovery. Reliability for resting HRV variables from our lab is as follows: SDNN: intraclass correlation coefficient (ICC) = 0.97, standard error of measure (SEM) = 6.4 ms; RMSSD: ICC = 0.70, SEM = 44.8 ms). 

### 2.5. Repeated Sprint Ability

Participants completed a repeated sprint test on a friction-loaded cycle ergometer (Monark 894E, Stockholm, Sweden). Following a two-minute warm-up cycling between 50 and 60 rpm against a resistance of 0.5 kg, a subsequent warm-up of two 30-s bouts of cycling between 85 and 115 rpm at a resistance of 1.5 kg was completed, with 60 s of passive rest in between. After the warm-up, participants completed 10 six-second maximal sprints at 65 g/kg body weight, with 30 s of passive recovery in between. Peak power (PP), fatigue index (FI), and average power (AP) were calculated by the manufacturer’s software. HR and rate of perceived exertion (RPE) per sprint were recorded. Reliability for AP and FI from our laboratory is as follows: (AP: ICC = 0.97, SEM = 120.8 W; FI: ICC = 0.97, SEM = 4.0%). 

### 2.6. Supplementation

Participants were randomly assigned to menstrual cycle start phase (follicular or luteal) and supplement type (creatine monohydrate or PL) using a computer-generated allocation sequence. Starting the day off or the day after their visit, participants were instructed to consume 20 g per day (4 × 5 g doses) of creatine monohydrate (Creapure^®^ AlzChem, Trostberg GmbH, Germany, GRAS Notice No. GRN 931 and with 2 g of Crystal Light) for five consecutive days or 4 × 5 g doses of a non-caloric powder for five consecutive days (20 g per day; Crystal Light). The creatine monohydrate and PL were identical in color and taste and were measured and packaged by an individual not directly involved in the distribution of the treatments. Both treatments were distributed in opaque containers. Participants consumed each individual 5 g dose with 6–8 ounces of water at regular intervals around the same time each day. Compliance was tracked with a log and returned with empty individual dosing packets at the end of each loading phase (creatine monohydrate: 96.6%; PL: 99.8%, compliance). A minimum of a four-week washout period was allotted before participants returned for their second randomized session. Participants continued their cycle tracking throughout the 4-week washout period. The supplement for the next cycle was distributed during the following pre-supplementation visit.

### 2.7. Dietary Intake

Participants were instructed to record their normal diet with as much detail as possible for two non-consecutive weekdays and one weekend day. Using nutrition analysis software (The Food Processor, version 10.12.0, Esha Research, Salem, OR, USA), diet logs were evaluated for average calories (CAL; kcal), carbohydrate (CHO; g), fat (FAT; g), protein (PRO; g), and relative protein (g/kg body mass) intake. When evaluated by paired sample *t*-tests, there were no significant differences between CAL (*p* = 0.987), CHO (*p* = 0.972), PRO (*p* = 0.395), FAT (*p* = 0.791), and relative protein intake (*p* = 0.670) between the PL and creatine monohydrate groups. 

### 2.8. Statistical Analysis

Separate ANCOVAs, covaried for fitness level from estimated VO_2_ max, were used to compare HRV values (SDNN, RMSSD [ms]) between supplement groups. Separate mixed factorial ANOVAs [2 × 2; treatment (creatine monohydrate vs. PL) × (Δ low hormone vs. Δ high hormone)] were used to compare the change scores between groups for all primary outcomes (AP [W], PP [W], tPP [ms], and FI [%]); in the event of a significant interaction, post hoc comparisons were analyzed using Bonferroni pairwise comparisons. For each sprint, change scores from baseline to post-supplementation were calculated for PP and TP. Change scores (post-pre) were calculated for total AP and FI in the low hormone vs. high hormone. A series of two-way (2 × 10; treatment × sprint) mixed model ANOVAs were used to compare change scores for each sprint between treatments. The Bonferroni method was used to evaluate post hoc comparisons. Analyses were performed using SPSS software (Version 27.0; IBM, Armonk, NY, USA). Statistical significance was set at 0.05. 

## 3. Results

### 3.1. Estrogen

There were no significant differences between levels of the low hormone and the high hormone (mean difference [MD]: 0.02 ± 0.37, *p* = 0.713). In the PL group, estrogen levels tended to be higher in the low hormone (0.87 pg/mL) compared to the high hormone (0.74 pg/mL), whereas in the creatine monohydrate group, values were higher in the high hormone (0.74 pg/mL) compared to the low hormone (0.69 pg/mL).

### 3.2. Heart Rate Variability (HRV)

#### 3.2.1. Standard Deviation of Normal-to-Normal RR Intervals (SDNN)

Average recovery change (0–15 min): There was no significant phase × supplement interaction (*p* = 0.552; η^2^: 0.10). In the PL group, values decreased in both the low hormone (Δ: −1.0 ± 2.0 ms) and the high hormone (Δ: −2.8 ± 2.9 ms) (Table 2); whereas the creatine monohydrate group displayed an increase in both the low hormone (Δ: 2.1 ± 2.0 ms) and the high hormone (Δ: 3.2 ± 3.0 ms). There was no main effect for phase (*p* = 0.355; η^2^: 0.024), with minimal change seen during the low hormone and the high hormone (Table 2). There was no main effect for the supplement (*p* = 0.095; η^2^: 0.075). There was a non-significant difference (Δ: −4.6 ± 1.8 ms) between the PL and creatine monohydrate groups. 

Immediate Recovery (first 5 min): There was no significant phase × supplement interaction (*p* = 0.293; η^2^: 0.034). In the PL group, there was an increase from low hormone-pre (mean ± SD: 77.3 ± 5.8 ms) to low post-hormone (80.0 ± 5.1 ms) and an increase from high pre-hormone (69.5 ± 4.9 ms) to high post-hormone (74.3 ± 4.7 ms). Whereas in the creatine monohydrate group, there was a decrease from low pre-hormone (78.6 ± 6.0 ms) to low post-hormone (74.1 ± 5.2 ms) and a decrease from high pre-hormone (65.7 ± 5.0 ms) to high post-hormone (59.8 ± 4.9 ms). There was no main effect for the phase (*p* = 0.382; η^2^: 0.025), with minimal differences seen between low pre-hormone and low post-hormone (Δ: 0.9 ± 3.8 ms) and between high pre-hormone and high post-hormone (Δ: 0.5 ± 4.1 ms). There was no main effect for a supplement (*p* = 0.299; η^2^: 0.030), but there was a small, non-significant difference between the PL and the creatine monohydrate group (Δ: 5.7 ± 5.4 ms). 

Resting: No significant differences were found between low pre-hormone and high pre-hormone (4.4 ± 34.8 ms; *p* = 0.439). There was no phase × supplement interaction (*p* = 0.352 η^2^: 0.030). There was a main effect for the phase (*p* = 0.030; η^2^: 0.079), with a decrease between low pre-hormone and low post-hormone (Δ: −2.4 ± 4.3 ms) and between high pre-hormone and high post-hormone (Δ: −6.6 ± 4.8 ms).

#### 3.2.2. Root Mean Square of Successive Differences (RMSSD)

Average recovery change (0–15 min): There was no phase × supplement interaction (*p* = 0.890; η^2^: 0.001). There was a greater decrease in the creatine monohydrate group from high pre-hormone to high post-hormone, compared to the PL group from high pre-hormone to high post-hormone (Table 2). There was no main effect for phase (*p* = 0.155; η^2^: 0.055). The greatest difference was seen between low post-hormone and high post-hormone (Δ: 2.5 ± 2.1 ms). There was no main effect for the supplement (*p* = 0.113 η^2^: 0.068). There was a non-significant difference between the PL and creatine monohydrate groups (Δ: −17.7 ± 10.9 ms). 

Immediate Recovery (first 5 min): There was no significant phase × supplement interaction (*p* = 0.542; η^2^: 0.010). There was no main effect for phase (*p* = 0.842; η^2^: 0.008). Low pre-hormone and low post-hormone values were higher compared to high pre-hormone and high post-hormone values (Table 2). There was no main effect for the supplement (*p* = 0.563: η^2^: 0.009). There was a small difference (Δ: 2.0 ± 3.4 ms) between the PL and creatine monohydrate groups. 

Resting: When pre-supplementation phases were compared, there were no significant differences (*p* = 0.104) between low pre-hormone and high pre-hormone. When compared across all four phases and by supplement group, there was no phase × supplement interaction (*p* = 0.331; η^2^: 0.031). Although close, there was no main effect for phase (*p* = 0.079; η^2^: 0.061). There was no main effect for the supplement (*p* = 0.566; η^2^: 0.009). There was a small, non-significant difference between the PL and creatine monohydrate groups (Δ: −5.4 ± 9.4 ms).

### 3.3. Exercise Performance

Total Average Power: There was no significant phase × supplement interaction (*p* = 0.293; η^2^: 0.030) for the change in AP across menstrual cycle phases (Table 3). There was no main effect for the phase (*p* = 0.406; η^2^: 0.019) or supplement group (*p* = 0.284; η^2^: 0.031). Though not significant, across both groups, the difference in AP was higher in the low hormone compared to the high hormone (Δ: 782.3 ± 930.9 W) and higher in the creatine monohydrate group compared to the PL group (Δ: 1002.0 ± 921.0 W).

Fatigue Index: For FI, there was a significant phase × supplement interaction (*p* = 0.048; η^2^: 0.101) (Figure 2). When decomposing the model, there were no significant differences across phases for the PL (*p* = 0.848) or creatine monohydrate (*p* = 0.143) groups. In the PL group, there was a slight decrease in the low hormone (Δ: −3.6 ± 5.9%) and a smaller change in the high hormone (Δ: 0.1 ± 8.1%). In the creatine monohydrate group, there was an increase in the low hormone (Δ: 4.1 ± 19.6%) and a decrease in the high hormone (Δ: −5.8 ± 19.0%) (Table 3). There was a significant main effect for a supplement (*p* = 0.048); the PL group exhibited a greater decrease in FI (Δ: −1.7 ± 2.2%) compared to the creatine monohydrate group (Δ: −0.8 ± 2.3%), although not significantly from each other (*p* = 0.781; η^2^: 0.002). FI resulted in an increase in the low hormone (Δ: 0.2 ± 2.3%) and a decrease in the high hormone (Δ:−2.8 ± 2.3%) (*p* = 0.366). 

Peak Power: There was no significant phase × supplement × sprint interaction (*p* = 0.498; η^2^: 0.035). There were no significant phase × supplement (*p* = 0.463; η^2^: 0.031), sprint × supplement (*p* = 0.681; η^2^: 0.026), or time × sprint interactions (*p* = 0.120; η^2^: 0.047). There was a main effect for the phase (*p* = 0.041; η^2^: 0.096) and sprint (*p* < 0.001; η^2^: 0.662). Pairwise comparisons for phase demonstrated low pre-hormone was significantly lower than high post-hormone (−19.2 ± 8.79 W; *p* = 0.039), and high post-hormone was significantly greater than high pre-hormone (22.9 ± 7.6 W, *p* = 0.005) (Table 2). Pairwise comparisons for sprints demonstrated a significant decrease in PP in sprints one through four compared to sprint 10 (*p* < 0.001) and for sprint five, compared to sprint 10 (*p* = 0.005). No significant differences were seen between bouts ten and six (*p* = 0.066) or bouts seven through nine (*p* = 0.999) (Figure 2).

## 4. Discussion

Previous research supports the potential for creatine supplementation to enhance recovery, through hydrogen ion buffering and pH regulation [4,10], increasing intramuscular PCr, [9,22,23] and promoting greater glycogen storage [24], demonstrated in males. Given physiological differences that could uniquely impact creatine utilization in female, such as creatine kinase activity across the menstrual cycle, there is potential for sex-specific differences in creatine monohydrate supplementation to influence exercise and recovery outcomes. In the current study, the luteal phase was associated with reduced HRV, sprint performance, and a greater fatigue index compared to the follicular phase. SDNN and RMSSD were lower, and FI was higher in the high hormone phase, suggesting delayed recovery and lower fatigue resistance. Creatine monohydrate supplementation had minimal effects on HRV and sprint performance outcomes, but current results suggest creatine monohydrate supplementation may have beneficial effects on FI. During the luteal phase, FI was significantly reduced with creatine monohydrate (−5.8 ± 19.0%), compared to the PL (0.1 ± 8.1%), suggesting greater fatigue resistance with creatine monohydrate supplementation.

### 4.1. Recovery from Exercise

To date, little data exists characterizing the response of SDNN and RMSDD over an exercise recovery period; however, it is known that HRV decreases in the five to thirty minutes following high-intensity exercise in men [14,25,26]. Previous research in men using a similar repeated sprint exercise stimulus resulted in a decrease in the natural logarithm of SDNN HRV immediately post-exercise, followed by a gradual rise after 10 min and 30 min [27]. In the present study, instead of a gradual increase, HRV remained elevated immediately after exercise (0–5 min), began to decrease by minute 10 of recovery, and then gradually increased at 15 min post-exercise, regardless of supplementation. Within the creatine monohydrate group, RMSSD values were lower compared to the PL group, however, this decrease may be explained by the greater increase in power output observed in the creatine monohydrate group (Table 2). While there was no significant phase × supplement interaction, average recovery (HRV) over 15 min in the PL group went down in both the low hormone (−1.0 ± 2.0 ms) and the high hormone (−2.8 ± 2.9 ms); whereas the creatine monohydrate group displayed an increase in RMSSD in both the low hormone (2.1 ± 2.0 ms) and the high hormone (3.2 ± 3.0 ms) from pre- to post-supplementation, suggesting creatine monohydrate may positively influence recovery rates across the entire menstrual cycle. Existing data is unclear on whether a specific menstrual phase effects performance or recovery differently, and to our knowledge, no data on post-exercise HRV exists in relation to the menstrual cycle phase. The present study suggests HRV may be lower in the high hormone, compared to the low hormone phase. Although not significant, results also suggest that creatine monohydrate supplementation provided a slight improvement in exercise recovery HRV values across both phases of the menstrual cycle. 

### 4.2. Resting HRV

There is limited data available evaluating the impact of creatine monohydrate on resting HRV, with none-to-date in females. One previous investigation in male bodybuilders reported attenuated time-domain values following four weeks of creatine monohydrate compared to placebo (MD: −15.9 ± 48.5 ms) [28]. The current study reported no significant effect of creatine monohydrate on resting HRV compared to PL (MD: −5.4 ± 9.4 ms). Since creatine is an energy buffer for short-term energy systems, utilized primarily in high-intensity exercise, it is unsurprising that it would not significantly impact resting values due to the difference in primary energy system usage. Data examining HRV at rest in different menstrual cycle phases suggests a higher sympathetic dominance at rest in the high hormone when compared to the low hormone, with a significant relationship to estradiol-17 and progesterone concentrations [16,29]. When healthy eumenorrheic females ages 18–25 years were evaluated during a five-minute rest period, both SDNN and RMSSD values were lower in the high hormone phase—compared to the low hormone phase. The present study resulted in similar effects, with the largest decrease seen in the high hormone phase (−6.6 ± 4.8 ms). Though not significant, the current study showed lower resting RMSSD values in the high hormone compared to the low hormone (−8.5 ± 32.0 ms). This data aligns with current literature suggesting a lower HRV at rest in the high hormone phase of the menstrual cycle, possibly due to high levels of estrogen and progesterone [16]. 

### 4.3. Exercise Performance

Short-term creatine supplementation is widely associated with improved anaerobic exercise. Creatine monohydrate supplementation has resulted in improvements in AP and PP over 15-s sprints in trained males [12]. A similar study in trained males examining sprint performance with a similar protocol of brief 10-s rest periods resulted in a significant increase in AP from pre- to post-supplementation in the creatine monohydrate group compared to the PL group. In a sample of female athletes, improvements in repeated sprint times were reported for the creatine monohydrate group from pre- to post-supplementation compared to the PL group [13]. Other anaerobic performance indices have been improved following creatine monohydrate supplementation in women [22,30,31,32]; anaerobic working capacity was significantly improved following creatine monohydrate loading compared to PL [33]. To date, besides our companion paper [34], we are unaware of any other studies that have accounted for the menstrual cycle with creatine monohydrate supplementation. In the current study, there was no significant increase in AP between groups, however, the creatine monohydrate group exhibited a higher AP compared to the PL group (MD: 1002.0 ± 921.0 W). This difference is well beyond the measurement error (120.8 W) and can be considered a real change. Non-significant improvements in PP were also demonstrated in the present study for the creatine monohydrate group compared to the PL group (Table 3), aligning with previous data in men and women supporting improvements in anaerobic performance following creatine monohydrate supplementation. Existing data examining anaerobic performance across the menstrual cycle, without supplementation, suggests no significant differences between menstrual cycle phases [35,36]. In the current study, there were no significant differences in AP between phases; however, AP was lower in the high hormone phase compared to the low hormone phase for both supplement groups. In summary, creatine monohydrate may improve sprint performance in women, with no effect of the menstrual phase.

Along with improved anaerobic performance, creatine monohydrate supplementation has been shown to delay fatigue due to its buffering capabilities. A study using a similar repeated sprint protocol in untrained males showed no significant differences in FI during repeated sprint exercise following creatine monohydrate supplementation from pre- to post-supplementation (MD: 2.68 ± 0.39%) [5]. In contrast, a study examining FI differences in different menstrual cycle phases reported the lowest FI in the low hormone phase compared to the midcycle and high hormone phases, with no statistical difference between phases [36]. In the current study, greater improvements were seen in the high hormone phase in the creatine monohydrate group (−5.8 ± 19.0%), compared to the PL group (0.1 ± 8.1%). Additionally, FI appeared to be impacted by the menstrual cycle, with a lower FI in the low hormone phase (0.2 ± 2.3%) compared to the high hormone phase (−2.8 ± 2.3%). The outcomes of this study suggest that an increase in fatigue, possibly seen in the high hormone phase, can be improved with short-term creatine monohydrate supplementation. 

The limitations of this study should be noted. This study used the counting method and cycle tracking from an app. This could result in inconsistencies in phase/day determination depending on participant adherence to daily tracking, which may have impacted accurate identification of menstrual cycle phase. It should be noted that participants were asked to track their menstrual cycle for at least one month prior to testing, and a familiarization session completed at the baseline visit was completed during menstruation/bleeding which allowed for greater tracking accuracy. The sample size was also smaller than anticipated for this study, which may have impacted statistical significance. Although post hoc sample analysis suggested that the study was adequately powered, the dropout rate in this study (*n* = 15; 28%) was higher than previous studies with female participants (10%) from our lab. Due to high rates of contraceptive use, women using select hormonal contraception (i.e., Monophasic oral contraception and intrauterine device), in addition to naturally cycling women were recruited and enrolled in order to gather pilot data on the impact of hormonal contraception on the presented outcomes. Due to the small sample sizes for each of those groups, additional stratification for statistical analyses was not conducted. To account for this, women were tested in similar hormonal phases, regardless of contraception type. The lack of significant differences in estrogen values highlights the difficulty of comparing menstrual cycles between individuals, considering the high variability and individuality. Further investigation should be done on progesterone levels, as well as examining the estrogen/progesterone ratio. 

## 5. Conclusions

There appeared to be a clinically meaningful effect of creatine monohydrate supplementation on some performance outcomes, particularly in the high hormone phase, as shown by a 5% decrease in FI in the creatine monohydrate group compared to a <1% change in the PL group. Though the findings were not statistically significant, the data suggests that creatine monohydrate could help counteract sprint performance decrements in the high hormone phase. This data can help inform creatine monohydrate loading strategies for active females. Creatine monohydrate supplementation did not appear to influence HRV values. Across both groups, resting HRV values were lower in the high hormone phase, suggesting the possibility of diminished recovery from exercise in this phase; this could also be attributed to the higher PP obtained during the cycling test in the creatine monohydrate group. More research in women is needed to examine the effect of a longer creatine monohydrate supplementation period on performance and recovery values across different hormonal profiles. Additionally, future research should evaluate the potential impact of creatine supplementation on amenorrheic females. It appears that a short-term loading phase may help reduce fatigue for women, particularly in their high hormone phase. The impact of hormonal contraception on these outcomes should be explored further. 

## Figures and Tables

**Figure 1 nutrients-15-03567-f001:**
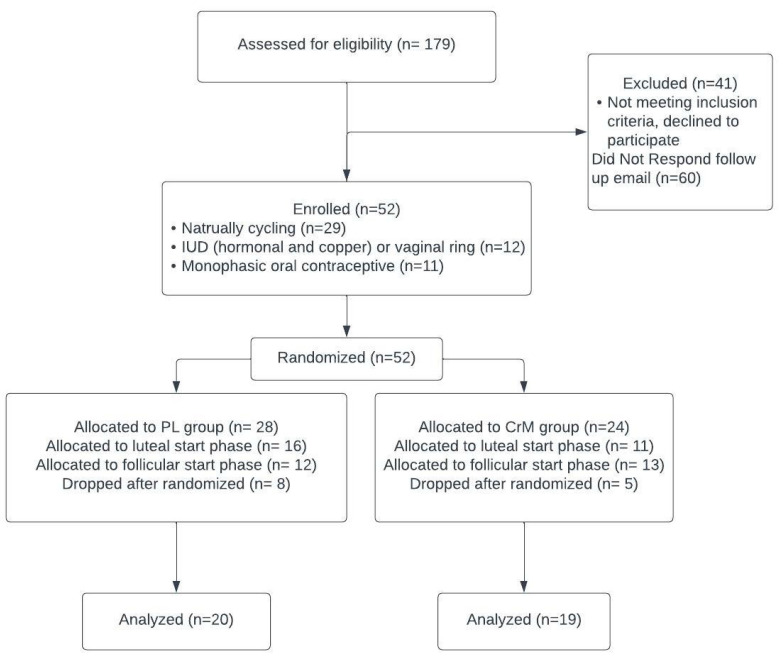
Consort diagram for recruitment and enrollment.

**Figure 2 nutrients-15-03567-f002:**
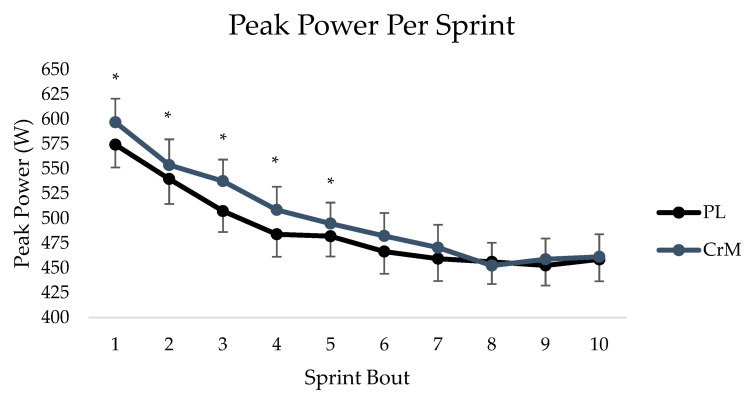
Peak power per sprint (W) across all sprints between placebo (PL—black) and creatine (CrM—gray) group. * indicates statistical significance for sprint trial compared to sprint 10.

**Table 1 nutrients-15-03567-t001:** Mean ± standard error demographics for creatine (CrM) and placebo (PL) group.

Supplement Group	Age (y)	Height (cm)	Weight (kg)	BMI (kg/m^2^)	Percent Body Fat (%)	Estimated VO_2_ Max (mL/kg/min)	Average Cycle Length (days)
CrM (*n* = 19)	25.5 ± 7.2	164.6 ± 6.1	66.2 ± 9.2	23.0 ± 5.0	26.9 ± 6.1	38.3 ± 7.3	28.9 ± 4.6
PL (*n* = 20)	23.8 ± 4.3	167.0 ± 4.8	64.1 ± 7.0	24.4 ± 2.9	27.9 ± 5.7	39.7 ± 4.6	28.1 ± 5.8

Birth control type: IUD or vaginal ring: (CrM *n* = 2, PL *n* = 4) oral contraceptive: (CrM *n* = 5, PL *n* = 3), natural cycle: (CrM *n* = 12, PL *n* = 13).

**Table 2 nutrients-15-03567-t002:** Mean ± Standard Error for standard deviation of normal-to-normal RR intervals (SDNN) and Root Mean Square of Successive Differences (RMSSD) at rest, average recovery (0-15 min) and immediate recovery (5 min Post exercise) for pre-and post-supplementation in the follicular (low hormone) and luteal (high hormone), separated by supplement group.

	Creatine (*n* = 19)				Placebo (*n* = 20)			
	*Follicular Pre*	*Follicular Post*	*Luteal Pre*	*Luteal Post*	*Follicular Pre*	*Follicular Post*	*Luteal Pre*	*Luteal Post*
SDNN								
Resting (ms)	120.6 ± 7.4	120.1 ± 7.4	112.6 ± 7.4	103.3 ± 7.8	114.6 ± 7.2	110.3 ± 7.2	113.7 ± 7.2	103.3 ± 7.8
Average Recovery (ms)	41.4 ± 3.0	39.4 ± 2.6	37.0 ± 0.0	33.7 ± 2.4	47.0 ± 2.9	48.0 ± 2.5	41.6 ± 2.9	44.4 ± 2.3
5 min Post-Exercise (ms)	78.0 ± 6.0	73.6 ± 5.2	65.1 ± 5.1	60.1 ± 4.8	77.9 ± 5.8	80.4 ± 5.1	70.0 ± 5.0	74.0 ± 4.7
RMSSD								
Resting (ms)	98.9 ± 7.3	93.1 ± 8.1	87.9 ± 8.1	75.5 ± 8.1	87.3 ± 7.1	84.6 ± 7.9	81.2 ± 7.9	80.5 ± 7.9
Average Recovery (ms)	12.2 ± 3.0	12.6 ± 2.7	12.0 ± 3.4	8.4 ± 2.1	16.1 ± 2.9	15.8 ± 2.6	15.6 ± 3.3	15.0 ± 2.0
5 min Post-Exercise (ms)	17.2 ± 4.0	16.3 ± 3.9	11.8 ± 3.2	9.4 ± 2.4	16.2 ± 3.8	15.6 ± 3.8	16.0 ± 3.1	14.9 ± 2.3

Data are presented as raw values. Significance was based on analyses of covariance (ANCOVA), covarying for an estimated VO_2_ max of 39.03 mL/kg/min.

**Table 3 nutrients-15-03567-t003:** Mean ± Standard Error for Total Average Power, Fatigue Index, Peak Power and Time to Peak Power for pre-and post-supplementation in follicular (low hormone) and luteal (high hormone), separated by supplement.

	Creatine (*n* = 19)				Placebo (*n* = 20)			
	*Follicular Pre*	*Follicular Post*	*Luteal Pre*	*Luteal Post*	*Follicular Pre*	*Follicular Post*	*Luteal Pre*	*Luteal Post*
Total Average Power (W)	3160.5 ± 246.5	5116.5 ± 5	2372.5 ± 319.0	2553.3 ± 336.8	2874.2 ± 241.2	2835 ± 1190.7	2845.4 ± 310.9	3017.2 ± 328.3
Fatigue Index(%)	29.2 ± 2.6	30.3 ± 2.6	37.9 ± 4.7	30.3 ± 3.5 *	27.5 ± 2.4	23.8 ± 2.6	27.6 ± 4.4	27.3 ± 3.3
Peak Power (W)	492.9 ± 21.9	508.1 ± 21.7	484.8 ± 25.0	520.6 ± 22.1	484.0 ± 21.2	488.8 ± 20.4	484.5 ± 24.1	494.6 ± 21.4
Time to Peak Power (ms)	1663.5 ± 181.6	2265.6 ± 207.3	1937.1 ± 159.7	2141.2 ± 239.0	1782.5 ± 175.5	2269.7 ± 200.2	2001.7 ± 154.3	2310.9 ± 230.9

* indicates statistical significance from pre- to post-supplementation (*p* ≤ 0.005).

## Data Availability

Data generated and analyzed from this study are available from the corresponding author upon reasonable request.

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
