# Peer review of "The Effects of Creatine Monohydrate Loading on Exercise Recovery in Active Women throughout the Menstrual Cycle"

_nutrients, 2023, doi:10.3390/nu15163567_

Round 1
Reviewer 1 Report
In the paper entitled “The Effects of Creatine Monohydrate Loading on Exercise Recovery in Active Women Throughout the Menstrual Cycle”, the authors evaluated the effect of creatine monohydrate loading on exercise recovery, measured from heart rate variability (HRV) and repeated sprint performance, in women across the menstrual cycle.
Despite the potentiality of being shared with the scientific community, I believe that the manuscript would benefit from a minor revision.
INTRODUCTION is too short - does not substantiate the theme, does not clarify fully the current level of knowledge - please review - please add significant details.
MATERIALS AND METHODS: More information should be provided about the participants’ characteristics
DISCUSSION AND CONCLUSIONS are not properly formatted.
Kind regards
Author Response
In the paper entitled “The Effects of Creatine Monohydrate Loading on Exercise Recovery in Active Women Throughout the Menstrual Cycle”, the authors evaluated the effect of creatine monohydrate loading on exercise recovery, measured from heart rate variability (HRV) and repeated sprint performance, in women across the menstrual cycle. Despite the potentiality of being shared with the scientific community, I believe that the manuscript would benefit from a minor revision.
Thank you to this reviewer for your support of our work. We have updated the introduction, description of the participants, as well as the discussion/conclusions. All changes are highlighted in yellow.
INTRODUCTION is too short - does not substantiate the theme, does not clarify fully the current level of knowledge - please review - please add significant details.
We have taken time to review and update the introduction to more clearly highlight that currently no data exists on evaluating exercise performance across the menstrual cycle with creatine supplementation (thus there is not a lot of literature to report on). Additionally, the introduction is currently at ~600 words, with scientific introductions suggested to be between 500-1000 words. We have added more details where possible including additional notes on potential mechanisms and current data on anaerobic performance.
MATERIALS AND METHODS: More information should be provided about the participants’ characteristics
Additional details have been added about the participants and were detailed in table 1. If there are elements that the reviewer would like to see, we would be interested to know specifics in order to add necessary details.
DISCUSSION AND CONCLUSIONS are not properly formatted.
We have reviewed the nutrients guidelines and formatting and have updated paragraphs to align. Additionally, we have reviewed this section to address any potential formatting concerns. If there was an element that the reviewer specifically noted, please share and we are more than willing to update.
Reviewer 2 Report
The manuscript of Gordon et al. describes the effects of creatine monohydrate supplementation on physical performance and recovery of women, taking into consideration their phases of the menstrual cycle. Although, no statistically significant changes were identified in the presence of creatine monohydrate, the study revealed interesting results that hold great potential in counteracting physical performance decrements in the luteal phase of female athletes. This is very important, because it seems that female athletes in the luteal phase are not able to provide the same physical performance as their competitors in the follicular phase. Therefore, pharmacological strategies to compensate for this transient hormonal “handicap” are desired. The study was meticulously prepared; numerous relevant parameters were taken into account. Reading the manuscript, I found only some minor issues that need to be addressed by the authors.
Minor issues
1. In Figure 2, it is shown peak power per sprint between PL and CrM groups. However, in the text (lines 285-286), the authors referred to this figure when evaluating phase x supplement interaction. This inconsistency should be addressed.
2. Line 96: Women with a BMI between 18.5-29.9 kg/m2 were included in testing. However, a BMI of 25-29.9 kg/m2 is considered overweight. The authors should discuss how or whether this could affect the outcomes of their study.
Author Response
Response to Reviewer 2
The manuscript of Gordon et al. describes the effects of creatine monohydrate supplementation on physical performance and recovery of women, taking into consideration their phases of the menstrual cycle. Although, no statistically significant changes were identified in the presence of creatine monohydrate, the study revealed interesting results that hold great potential in counteracting physical performance decrements in the luteal phase of female athletes. This is very important, because it seems that female athletes in the luteal phase are not able to provide the same physical performance as their competitors in the follicular phase. Therefore, pharmacological strategies to compensate for this transient hormonal “handicap” are desired. The study was meticulously prepared; numerous relevant parameters were taken into account. Reading the manuscript, I found only some minor issues that need to be addressed by the authors.
Thank you for the kind words and added review. We appreciate the attention to detail and have addressed the comments below.
Minor issues
- In Figure 2, it is shown peak power per sprint between PL and CrM groups. However, in the text (lines 285-286), the authors referred to this figure when evaluating phase x supplement interaction. This inconsistency should be addressed.
Thank you for this viewpoint. The description of statistical significance has been updated to reflect the significance across sprint trial and not between supplement groups.
- Line 96: Women with a BMI between 18.5-29.9 kg/m2 were included in testing. However, a BMI of 25-29.9 kg/m2 is considered overweight. The authors should discuss how or whether this could affect the outcomes of their study.
Thank you for this thoughtful question. The inclusion criteria was set up to 29.9 as to remove already stringent inclusion criteria. Based on the mean BMI values, along with percent body fat, these women were actually normal weight and quite lean (per DXA NHANES values). BMI values for each group have been added to Table 1, along with a note regarding their stature.